# The Dual Function of PhOH Included in the Coordination Sphere of the Nickel Complexes in the Processes of Oxidation with Dioxygen

**DOI:** 10.3390/molecules27113502

**Published:** 2022-05-30

**Authors:** Matienko Ludmila, Zhigacheva Irina, Mil Elena, Albantova Anastasia, Goloshchapov Alexander

**Affiliations:** Emanuel Institute of Biochemical Physics, Russian Academy of Science, 4 Kosygin Street, 119334 Moscow, Russia; zhigacheva@mail.ru (Z.I.); elenamil2004@mail.ru (M.E.); albantovaaa@mail.ru (A.A.); golan@sky.chph.ras.ru (G.A.)

**Keywords:** catalysis, etylbenzene oxidation, nickel complexes, PhOH, H-bonds, supramolecular structures, AFM-method

## Abstract

The role of ligands in the regulation of the catalytic activity of Ni-complexes (Ni(acac)_2_) in green process-selective ethylbenzene oxidation with O_2_ into α-phenyl ethyl hydroperoxide is considered in this article. The dual function of phenol (PhOH) included in the coordination sphere of the nickel complex as an antioxidant or catalyst depends on the ligand environment of the metal. The role of intermolecular H-bonds and supramolecular structures (AFM method) in the mechanisms of selective catalysis by nickel complexes in chemical and biological oxidation reactions is analyzed.

## 1. Introduction

Currently, antioxidants (AO) are widely used to stabilize different oxidation processes: the oxidation of hydrocarbons, fuels, lubricating oils, plastics, and industrial rubber goods. Emanuel and his students determined the mechanism of action of antioxidants during the oxidation of organic compounds, and gave them a definition as compounds that inhibit the development of free radical oxidation processes. These studies, among other things, led to the creation by Emanuel of the theory of free radical chain reactions in the liquid phase [1,2].

In medicine and the food industry, natural, non-toxic antioxidants were initially used: α-tocopherol, cysteine, glutathione, and thiourea. Later, artificial, synthetic, fat-soluble antioxidants—aromatic phenols—were tested. Dibunol was the first drug in this group of antioxidants. Ionol and dibunol were researched as antioxidants for the study of the mechanism of liquid-phased hydrocarbon oxidations. In medicine, dibunol has proven itself well in the treatment of burns, bladder cancer, ulcerative lesions of the skin, and mucous membranes. Another phenolic antioxidant, probucol, is effective in the prevention of atherosclerosis [3,4].

According to their chemical properties, antioxidants (AO) are divided into two groups: “radical traps” that directly interact with free radicals, and “scavengers” (cleaners) that decompose the products of free radical oxidation (FRO) with their subsequent inactivation and utilization [2].

Like the radicals that are considered prooxidants, surprisingly, antioxidants can also have prooxidant properties [5]. Vitamin C is considered a potent antioxidant and intervenes in many physiological reactions, but it can also become a prooxidant. This happens when it combines with iron and copper, reducing Fe^3+^ to Fe^2+^ (or Cu^3+^ to Cu^2+^), which in turn reduces hydrogen peroxide to hydroxyl radicals [6]. a-Tocopherol is also known to be a useful and powerful antioxidant, but in high concentrations it can become a prooxidant due to its antioxidant mechanism. When it reacts with a free radical it becomes a radical itself, and if there is not enough ascorbic acid for its regeneration it will remain in this highly reactive state and promote the autoxidation of linoleic acid [5]. Although not much evidence has been found, it is proposed that carotenoids can also display prooxidant effects, especially through autoxidation in the presence of high concentrations of oxygen-forming hydroxyl radicals [5]. Flavonoids can act as prooxidants, although each one responds differently to the environment into which it is inserted. Dietary phenolics can also act as prooxidants in systems that contain redox-active metals.The presence of O_2_ and transition metals like iron and copper catalyze the redox cycling of phenolics, and may lead to the formation of ROS and phenoxyl radicals which damage DNA, lipids and other biological molecules [7].

This article discusses the double phenomenal function of phenol, included in the coordinate sphere of nickel complexes, as an antioxidant or catalyst in the oxidation of ethylbenzene O_2_. The role of hydrogen bonds and supramolecular structures, based on nickel complexes with PhOH as models of enzymes, are also analyzed.

The detailed establishment of the mechanism of the catalytic oxidation of ethylbenzene made it possible to propose methods for controlling the process, with the aim of directed oxidation to the corresponding hydroperoxide. The problem is of not only theoretical but also practical interest, as ethylbenzene hydroperoxide is used as an intermediate product in multi-ton production and the joint production of propylene oxide and styrene.

## 2. Results and Discussion

The chemical industry occupies one of the first places among the sources of hazardous environmental pollution; in this regard, it is in this area in which it is especially necessary to search for fundamentally new chemical processes characterized by a low level of energy consumption and the minimal formation of byproducts. The development of industrial processes for the oxidation of hydrocarbons is determined by the ability of researchers to control these processes. An effective method of controlling the rate and mechanism of the free radical autoxidation of hydrocarbons is the use of a catalyst. Based on the study of the mechanism of catalysis by nickel complexes in the oxidation of alkylarenes (ethylbenzene, cumene), Matienko [8] was the first to propose a method for the modification of homogeneous catalysts by adding electron-donating ligands in order to increase the efficiency of the catalysts.

Catalyst deactivation processes always accompany the catalytic oxidation of hydrocarbons. Matienko established a mechanism for the deactivation of nickel compounds Ni(acac)_2_ (or NiSt_2_) during the oxidation of ethylbenzene. Nickel catalysts are not simply deactivated but turn into antioxidants due to the formation of complexes with phenol, which is one of the oxidation products of ethylbenzene. Under the conditions of the radical chain reaction of ethylbenzene oxidation, the resulting Ni(acac)_2_•PhOH complexes have an antioxidant effect on ethylbenzene oxidation in two ways: they decompose ROOH (α-phenyl ethyl hydroperoxide (PEH) heterolytically with the formation of phenol and acetaldehyde, and also react with free radicals, RO_2_•, to terminate the oxidation chains [9]. At the same time, mixtures of Ni(acac)_2_ with phenols (simple or 4-tert-butylphenol) are effective inhibitors of the oxidation of hydrocarbons of various classes—alkylarenes and alkanes (USSR Author’s Certificate No. 530875 (Matienko, Maizus)).

Based on the calculations (Matienko, Brin, Mosolova, Maizus), a quantitative confirmation of the previously established experimental facts was obtained that at the earliest stages of oxidation, at very low concentrations of phenol, Ni(acac)_2_•PhOH complexes are formed in the system that decompose PEH heterolytically and have the ability to inhibit the oxidation process [9]. The following Figure 1 of PEH heterolysis under the influence of the complex Ni(acac)_2_•PhOH was proposed [9]:

Based on the studied mechanism of ethylbenzene oxidation, catalyzed by Ni(acac)_2_, a method of increasing the efficiency of the catalytic process was proposed. This method of modifying homogeneous catalysts with the addition of electron-donor ligands (Matienko) [8] made it possible not only to inhibit the formation of phenol but also to prevent the complex {Ni(acac)_2_•PhOH} formation. This method also made it possible to significantly increase the activity of nickel complexes as catalysts for the selective oxidation of alkylarenes to hydro peroxides, which are semi-products of large-scale industries (propylene oxide and styrene (ethylbenzene), phenol and acetone (cumene)). The most effective binary systems as catalysts for the oxidation of ethylbenzene in ROOH are {NiL^1^_n_ + L^2^} (L^1^ = acac, enamac ions, L^2^ = crown ethers, quaternary ammonium salts). The mechanism of their action is associated with the formation during the reaction of active primary complexes (NiL^1^_2_)_x_(L^2^)_y_ (I macro-stage of oxidation) and heteroligand complexes (Ni_x_L^1^_y_(L^1^_ox_)_z_(L^2^)_n_ (L^1^_ox_ = (OAc)^−^ion), which are intermediate products of the oxidation of primary complexes (NiL^1^_2_)_x_(L^2^)_y_ by molecular O_2_ (macro stage II of oxidation). We found that the mechanism of transformation of nickel complexes (NiL^1^_2_)_x_(L^2^)_y_ by molecular O_2_ (with the formation of the (OAc^−^) ligand, acetaldehyde and CO, and the subsequent formation of Ni_x_L^1^_y_(L^1^_ox_)_z_(L^2^)_n_) is similar to the mechanism of action of Ni-ARD (Acireductone Dioxygenase) and Cu(II), Fe(II)-containing Quercetin 2,3-Dioxygenases [10].

The phenomenal effect of increasing the catalytic activity of binary systems was discovered by L.I. Matienko and L.A. Mosolova in the case of introducing a third component, phenol, into the catalytic system {Ni(acac)_2_ + L^2^}. In the presence of the three-component system {Ni(acac)_2_ + L^2^ + PhOH}, a synergistic increase in the parameters ***C*** (the degree of conversion at ***S***_PEH_ ~ 90%) and the maximum concentration of hydroperoxide [PEH]_max_ were observed compared with catalysis by the binary system {Ni(acac)_2_ + L^2^} (L^2^ = monodentate ligands MSt (M = Na, Li), N-methyl-2-pyrrolidone (NMP), HMPA, DMF. (RF Patent 2004, authors Matienko and Mosolova. Patentee IBCP RAS). The synergistic effects of an increase in ***C***, and [PEH]_max_, upon the introduction of phenol into the {Ni(acac)_2_ + L^2^} catalytic system indicated an unusual catalytic activity of the resulting {Ni(acac)• L^2^ • PhOH} triple complexes [8].

This work presents new kinetic data obtained by GLC with the use of the Mathcad and Graph2Digit computer programs for the processing of experimental data, for the case of ethylbenzene oxidation catalyzed by two triple systems {Ni(acac)_2_ + L^2^ + PhOH} (L^2^ = NaSt) (Figure 1, Figure 2 and Figure 3).

The constancy of the reaction rate in the oxidation of ethylbenzene, catalyzed by the three-component system {Ni(acac)_2_ + L^2^ + PhOH} (Figure 1) for a long time t > 20 h (L^2^ = HMPA) t > 30 h (L^2^ = NaSt), indicates the extreme stability of the resulting triple {Ni(acac)_2_ ∙ L^2^ ∙ PhOH} complexes. The mechanism of the formation of ethylbenzene oxidation products catalyzed by {Ni(acac)_2_ • L^2^ • PhOH} complexes was evaluated by the method of graphical differentiation. As can be seen from Figure 2a, the ratio of the rates of formation of the reaction products AP and PEH is nonzero at t → 0: *w*_AP_ / *w*_PEH_ ≠ 0 at t → 0. Analogy dependence was received for MPC: *w*_MPC_/*w*_PEH_ ≠ 0 at t → 0 (cat = {Ni(acac)_2_ • NaSt • PhOH}). Similar regularities have been established for catalysis by systems including L^2^ = HMPA, and NMP. As such, the byproducts acetophenone and methylphenylcarbinol are not formed during the decomposition of PEH, but in parallel with PEH, apparently, in the reactions of chain propagation (Cat + RO_2_˙→) and quadratic chain termination (RO_2_˙ + RO_2_˙→). A “latent-radical” mechanism for the formation of products through the stage of chain continuation (Cat + RO_2_˙), including the homolytic decomposition of the intermediate {ROO-Cat} complex at the O–O bond, was proposed as a probable one (Figure 2) [8,11,12]. In the process of ethylbenzene oxidation, catalyzed by the triple systems {Ni(acac)_2_•HMPA(NMP)•PhOH}, the parallel formation of AP and MPC was also observed: *w*_AP_/*w*_MPC_ ≠ 0 at t → 0. However, in the case of catalysis by the {Ni(acac)_2_ • NaSt • PhOH} complexes, the sequential formation of AP from MPC was observed: *w*_AP_/*w*_MPC_ → 0 at t → 0 (Figure 2b).

As one can see in Figure 3, the concentration of phenol at first decreases sharply, then practically does not change for a significant period, and then slowly increases (t ≤ 30 h). In this case, the rate of phenol accumulation is slower than in the reaction catalyzed by the binary system {Ni(acac)_2_ + NaSt}.

The consumption of phenol is apparently due to the formation of triple complexes and, to a lesser extent, inhibitory activity. It was found that the rate of consumption of phenol practically does not change with an increase in the concentration of the ligand L^2^ by an order of magnitude during catalysis by triple systems {Ni(acac)_2_∙L^2^∙PhOH}. Earlier, a significant increase in the rate of formation of free radicals in the oxidation of ethylbenzene catalyzed by Ni(acac)_2_ in the presence of L^2^ (NaSt, NMP) was established due to the participation of the formation of more active complexes Ni(acac)_2_∙L^2^ in the stages of the chain initiation and radical decomposition of PEH [8,9].

In the case of catalysis by nickel complexes Ni(acac)_2_∙L^2^ in the absence of PhOH, the formation of ROOH (PEH), MPC, and AP at the stages of the chain initiation, decomposition of PEH, and chain termination—based on our own and literature data—can be represented as in Figure 3 (reactions 1–5) [8]. PhOH formation at catalysis by Ni(AcO)_2_ at deep oxidation stages is represented by reaction 6 (Figure 3) [8].

We assumed that the high stability of the complexes {Ni(acac)_2_∙L^2^∙PhOH} during the oxidation of ethylbenzene was one of the reasons which also may be associated with the formation of intermolecular bonds, i.e., H-bonds (phenol-carboxylate) and, possibly, other non-covalent interactions, leading to the formation of stable supramolecular structures. In favor of the formation of supramolecular structures based on triple complexes {Ni(acac)_2_∙L^2^∙L^3^} (L^3^ = PhOH) in the real conditions of catalyzed oxidation is the AFM data. The spontaneous process of self-organization of the studied triple complexes {Ni(acac)_2_∙L^2^∙L^3^} into a stable macrostructures on the modified silicon surface is due to the balance between intermolecular and molecular-surface interactions, which can be a consequence of hydrogen bonds and other non-covalent interactions (Figure 4a–c) [13,14,15].

Using the AFM method, we have shown for the first time the possibility of the self-organization of nickel complexes{Ni(acac)_2_∙L^2^∙ L^3^} (L^3^ = PhOH), which are not only effective catalysts of ethylbenzene oxidation but also structural and functional models of Ni-ARD (Ni-Acireductone Dioxygenase) [10], into supramolecular structures due to intermolecular H-bonds. UV spectroscopy data indicate the intra- and outer-sphere coordination of the extra-ligands L^2^ and L^3^ with M(acac)_n_ (M = Ni, Fe, n = 2,3, L^2^ = NMP, His, L^3^ = Tyr) [15,16].

Ni(Fe)ARD (Acireductone Dioxygenases) are involved in the methionine utilization pathway (MSP), which is a versatile pathway for the conversion of sulfur-containing metabolites to methionine. Ni(Fe)ARDs are an unusual case of catalysis. Depending on the nature of the metal ion in the active site, nickel–iron enzymes differ in their mechanism of action. They catalyze the conversion of the same substrates (1,2-dihydroxy-3-oxo-5(methyl thio)pent-1-ene (β-diketone, Acireductone) and molecular oxygen) into different products. FeARD catalyzes the penultimate step of the metabolic pathway for the oxidation of Acireductone to formate and 2-keto-4-(thiomethyl)butyrate, which is a precursor of methionine. The reaction pathway catalyzed by NiARD does not produce methionine, but this reaction produces CO, which is a neurotransmitter. CO has been identified as an anti-apoptotic molecule in mammals. Recently, a human enzyme was found to regulate the activity of matrix metalloproteinase I (MMP-I), which is involved in tumor metastasis, by binding to the cytoplasmic transmembrane tail peptide MMP-I [16,17].

Convincing evidence has been obtained in favor of the involvement of the tyrosine fragment in the stabilization of primary Ni-complexes in the mechanism of Ni-ARD action, as one of the regulatory factors reducing the activity of Ni-ARD. Thus, we observed the formation of very stable supramolecular structures based on model Ni-ARD triple systems, {Ni(acac)_2_∙L^2^∙L^3^} (L^2^ = NaSt, HMPA, His (L-Histidine), L^3^ = PhOH, Tyr (L-Tyrosine) (Figure 4a–d).

Similarly, the self-organization into supramolecular nanostructures observed in the case of model metal complexes of porphyrin with amino acids, tyrosine, and histidine can be considered as a regulatory factor influencing the oxygenation reactions of organic substances going through the mechanisms of action of Cytochrome P450-dependent monooxygenases [17,18,19].

Thus, the advantage of triple complexes {Ni(acac)_2_∙L^2^∙PhOH} as catalysts is their higher efficiency, as result of the absence of oxidative transformation during the reaction. The coordination of phenol apparently prevents the reaction of O_2_ inclusion into the acac^−^ ligand and the following oxidation transformation of the nickel complex. The high stability of complexes {Ni(acac)_2_∙L^2^∙PhOH} during the oxidation of ethylbenzene can also be associated with the formation of intermolecular bonds, H-bonds (phenol carboxylate) and, possibly, other non-covalent interactions leading to the formation of stable supramolecular structures. The AFM data provided evidence in favor of the formation of supramolecular structures based on triple-nickel complexes, including PhOH, under real conditions of catalyzed oxidation. Using AFM on model systems, convincing evidence has been obtained for the participation of a tyrosine fragment, located in the second coordination sphere, and intermolecular H-bonds in the stabilization of primary Ni-ARD complexes. This may be one of the reasons for the regulatory action of Tyr, which may reduce the activity of the Ni-ARD enzyme. The role of the second coordination sphere, including the Tyr fragment, and the participation of the Tyr fragment in the decrease in the enzyme activity is suggested, for example, in the action of Homoprotocatechuate 2,3-dioxygenase [17].

## 3. Materials and Methods

Ethylbenzene (RH= C_6_H_5_CH_2_CH_3_) was oxidized with dioxygen at 120 °C in a glass bubbling-type reactor in the presence of catalytic systems. This was followed by the analysis of the oxidation products. *α*-Phenylethylhydroperoxide (PEH = C_6_H_5_HCOOH(CH_3_)) was analyzed by iodometry. Thebyproducts, including methylphenylcarbinol (MPC = C_6_H_5_HCOH(CH_3_)), acetophenone (AP = C_6_H_5_CO(CH_3_)), and phenol (PhOH = C_6_H_5_OH), as well as the RH content in the oxidation process, were examined by GLC [8]. The order in which PEH, AP, and MPC formed was determined from the time dependence of the product accumulation rate rations at t → 0. The variation of these rations with time was evaluated by graphic differentiation [8]. The experimental data processing was performed using the special computer programs Mathcad and Graph2Digit.

In the AFM study, we used the scanning probe microscope SOLVER P47 SMENA10 (Moscow Adm. Distr. Zelenograd Russia) at a frequency of 150 kHz, using an NSG30 cantilever with a radius of curvature of 10 nm. We used an NSG30_SS cantilever (Nanosensors^TM^ Advanced Tec^TM^ AFM probes, Neuchatel, Switzerland) with a radius of curvature of 2 nm, a resonance frequency of 300 kHz, and a force constant of 22–100 N/m. TipsNano in tapping mode was used for the AFM research of supramolecular structures. Sampling was carried out using a spin coating process, from a CHCl_3_ solution (Ni(acac)_2_∙L^2^∙PhOH (Tyr) complexes). The measurement was conducted on air-dried samples. 

AFM SOLVER P47/SMENA/ was used with Silicon Cantilever NSG11S (NT MDT), which has curvature radius of 10 nm, or a 2-nm tip height, 10–15 µm, and a cone angle ≤ 22°. It was used in taping mode at a resonant frequency of 150 KHz [10,17]. A waterproof modified Silicon surface was used for the self-assembly growth due to H-bonding of Ni-contaning systems. We used the method of UV-spectroscopy first to prove the role of the His-fragment in the formation of nickel complexes. Quartz cuvettes, 1 mm thick, were used to record the spectra in the UV regions. The spectra were recorded on a high-sensitivity spectrophotometer, “UV-VIS-SPECS”, with UV-VIS-Analyst software.

## 4. Conclusions

The dual function of phenol, coordinating with the nickel complex, depending on the ligand environment of the metal (nickel) was discovered. Complexes {Ni(acac)_2_∙PhOH}-are efficient antioxidants in the oxidation of ethylbenzene with O_2_, and triple complexes {Ni(acac)_2_∙L^2^∙PhOH} are effective catalysts for the selective oxidation of ethylbenzene in PEH. The self-organization of stable supramolecular structures based on model triple-nickel complexes (AFM) is evidence in favor of the regulatory role of the Tyr fragment in the mechanism of action of the enzymes of Ni(Fe)ARD Dioxygenases.

## Data Availability

Not applicable.

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
