# Peer review of "The Dual Function of PhOH Included in the Coordination Sphere of the Nickel Complexes in the Processes of Oxidation with Dioxygen"

_molecules, 2022, doi:10.3390/molecules27113502_

Round 1

Reviewer 1 Report

Manuscript has been significantly improved following my previous considerations. However, introduction should also highlight the importance of the choice of ethylbenzene as substrate and of the selectivity in the oxidation products.

Moreover schemes reporting chemical structures or reactions need to be re-designed in an appropiate way

Author Response

We have made changes in accordance with the reviewer's comments. The last paragraph has been added in the Introduction, revealing the meaning of using ethylbenzene as an object of research. In addition, changes have been made to the Schemes 1-3. Structural formulas and decoding of the notation used are given, including in the captions to the Schemes 1-3.

Reviewer 2 Report

Topic, Phoh should be PhOH

Introduction, Fe3+ to Fe2+ also Cu,  charge should be superscript

Scheme 1: maybe show interaction between PhOH and OH of hydroperoxide 

Scheme 3: use chemical structure of ethylbenzene and other by-product instated of RH, AP and etc.

It would be good  to present activity of each complexes with products and its ratio in a table. So, reader could see the overall results. 

Author Response

  • Everything is right. I did not find PhOh.
  • In Introduction:

Page 1, line 4 below Fe3+ to Fe2+ (or Cu3+ to Cu2+) fixed on Fe3+ to Fe2+ (or Cu3+ to Cu2+ , marked pink.

  • Introduction, Page 2, last Paragraph – inserted, marked pink.
  • In Results and Discussion:

Page 2, Par.1, line 3 below: a correction has been made – modifying, marked pink.

Page 3, Scheme 1:

  1. PhOH is replaced by C6H5OH
  2. In complex Ni(acac)2C6H5O(-)•••H(+) the shift density at phenol coordination is shown. Interaction with ROOH and heterolysis of ROOH in a non-aqueous media presumably occurs through the formation of an intermediate complex (see Scheme 1).

 All changes marked pink.

Page 5. Scheme 2 and captions to Scheme 2: inserted transcripts of the radical R and R’. All changes marked pink.

Page 7. Scheme 3 and captions to Scheme 3: inserted transcripts of RH, PEH, AP, MPC, the radical R, R’, R’’. All changes marked pink.

      Page 9. In Materials and Methods:

     The transcripts of RH, PEH, MPC, AP, PhOH. All changes marked pink.

Comment: It would be good  to present activity of each complexes with products and its ratio in a table. So, reader could see the overall results.

Thank you for this comment.

The fact is that in this article, we have focused on dual function of PhOH in coordination sphere of nickel catalyst, on stability of triple complexes, including PhOH as catalysts, their resistance to oxidative decompositions, what is expressed in the non-reduction of the rate of accumulation of products for a long time. We provided data indicating in favor of the formation of stable supramolecular structures, based on triple complexes, due to intermolecular H-bonds. We will take into account your remark to evaluate the activity of triple complexes in the future in our works.

This manuscript is a resubmission of an earlier submission. The following is a list of the peer review reports and author responses from that submission.

Round 1

Reviewer 1 Report

According to me, the manuscript in the present form cannot be accepted.

First, especially on the first part, there is a significant lack of novelty while describing the role of phenols in modulating oxidative stress. In particular here authors investigate just one molecule (2-ethyl-6-methyl-3-hydroxypyridine). Not different derivatives, but the same molecule mixed with two different compounds. Moreover, this molecule in the mentioned formulations is already a commercial product.

The first part has no novelty.

The secnod part could be of interest, but the target of the whole paper should be completely changed, focusing just on catalysis, providing an appropriate introduction with references related to the oxidation of ethylbenzene, and better investigating the role of phenols as pro-oxidants.

Reviewer 2 Report

Introduction does not clear about significant of the work. 

Introduction part would be dementated relationship between antioxidant and/or phenolic compounds and nickel. In this research area, what have been done before and then what is important of your research. 

Schemes 1-2 must be fixed:  Structure of Ac-Hp and Ethoxidol can be in the same figure and assign number to the compounds, use the term figure instead of scheme. 

Section 2.2 : It would be good if authors could add a scheme to show oxidation reaction of ethylbenzene to give several products including peroxide and phenol.